# OpenReview forum: "StereoMap: Quantifying the Awareness of Human-like Stereotypes in Large Language Models"
_EMNLP/2023/Conference — EMNLP 2023 Main_

### Official Review · Reviewer_5TSk · 2023-07-27

**Soundness:** 3

**Excitement:**

3: Ambivalent: It has merits (e.g., it reports state-of-the-art results, the idea is nice), but there are key weaknesses (e.g., it describes incremental work), and it can significantly benefit from another round of revision. However, I won't object to accepting it if my co-reviewers champion it.

**Missing References:**

1. Reference for Cao et al. 2022 should be moved to the main body since part of the science is situating the work in context.

2. A related reference could be added: Herold et al. 2022 (Applying the Stereotype Content Model to assess disability bias in
popular pre-trained NLP models underlying AI-based assistive technologies)

**Paper Topic And Main Contributions:**

The authors propose a framework to assess stereotypes in LLM. The framework is based on the Stereotype Content Model (SCM) from social psychology. It contains three steps: ratings for SCM, keywords extraction and analysis, and the reasoning analysis of the economic status.

**Questions For The Authors:**

1.  Lines 25-27, how do you show that LLMs often rely on research findings to support their reasoning? (Did you check that the findings that models mention are from the real research papers and not something else?)
2. How do you see an extension of your work: to consider other social groups or models?
3. How many generations are done per social group/trait combination in order to get values for warmth and competence?
4. Figure 5 Appendix - are these the only groups for which models refuse to answer? Specify how many attempts were done per group.

**Reasons To Accept:**

Study of stereotypes in LLMs grounded in already existent and tested model from social psychology.  Proposes different ways to discover stereotypes in the models without human annotators. These measures are easy to utilize.

**Reasons To Reject:**

My concern is that discovering stereotypes in LLM by asking the model itself about how stereotyped it is is a bit unreliable and biased. Even though, this is done implicitly through SCM traits. Chat GPT (and overall models from that family) tend to block generations about marginalized social groups, so the question is how reliable these scores are?

I also see some unexpected results. For instance, from Figure 2 we can see that Blind people have a pretty high competence level (higher than blue-collar workers, for example) and retarded people have higher competence than young people. etc.  How would you explain this? Basically, the distribution of the traits is likely skewed.

Does this direct way to ask models about traits give us reliable information? Maybe it means just undiscovered bias that would show up in another way?

I am not convinced that the full stereotypes assessment of models' results from these initially biased models is enough and that it could go without human annotations. As an initial step - yes, but since the model distorts certain representations, there should be involved additional resources.

Also, please, see the questions and address the missing reference.

**Reproducibility:**

1: Could not reproduce the results here no matter how hard they tried.

**Reviewer Confidence:**

5: Positive that my evaluation is correct. I read the paper very carefully and I am very familiar with related work.

**Typos Grammar Style And Presentation Improvements:**

Line 20. -"Perceptions towards these social groups", sounds confusing, which social groups? they are not introduced in the abstract before
Line 266. - refers to the wrong table
Line 363 - Figure 2 refers to the Table 2

---

> ### Author Rebuttal · Authors · 2023-08-28
>
> We deeply appreciate your thoughtful feedback on our work. We will be happy to engage in follow-up discussion through the open review if the reviewer wishes!
>
> We would like to outline the distinctive characteristics of the task at hand. In contrast to tasks that possess a discernible ‘ground truth’, such as mathematical problem-solving or question-answering tasks, where prompting reliability is often gauged by quantifiable metrics like accuracy, the focal task in our study lacks a straightforward ‘ground truth’. Instead, our interest lies in comprehending the behavior of Large Language Models (LLMs) under specific conditions. This intrinsic complexity is one of the key reasons behind our reliance on well-established literature in the realm of social sciences. Rather than devising potentially subjective questions ourselves, we draw upon existing studies that offer robust insights into the very nature of stereotypes and their evaluative methodologies. By embracing the queries and theories that social scientists have historically employed to elicit responses from human subjects, we aim to demonstrate the alignment or divergence between LLMs and human behavior regarding the manifestations of stereotypes.
>
> It is imperative to underscore that one of the core objectives of our study is to illustrate how LLMs mirror or diverge from human tendencies in exhibiting stereotypes. In light of this objective and the inherent constraints of the task, we believe that our approach was situated in the relevant literature. Nonetheless, we acknowledge that the validation of our chosen framework might have been underemphasized in our paper. To address this concern, we have conducted an additional ablation analysis to underscore the sensitivity of our approach to various prompts. Furthermore, we conducted human evaluations of model outputs to evaluate their coherence and consistency under this framework. (To prevent the text from being verbose and to keep it concise, we’d like to direct you to the table which we have addressed under the rebuttal of reviewer NHSL for the details if you are interested.) The intersection of language models and social science inherently presents intricate challenges, and our research strives to bridge this gap by conscientiously adapting established social science paradigms.
>
> - The observed disparity in competence scores between ‘Blind people’ and ‘Blue collar workers’. We conducted an in-depth review, cross-referencing both the raw and processed data, and we affirm the precision of the provided values. Indeed, the mean competence score for ‘Blind people’ stands at 3.328, while for ‘Blue-collar workers’, it registers at 3.25.
>
>  We understand that such disparities may seem counterintuitive at first glance. We believe that these nuances could be attributed to the models' underlying mechanisms (or intentionally) designed to mitigate potential stereotype-related harm. For instance, when we explore the reasons assigned by DAVINCI for the aforementioned ratings, we discover that for 'blind people,' the model cites qualities like 'overcoming obstacles' and 'perseverance.' Again, one of the objectives of this paper is to scrutinize and probe the model behavior itself, rather than eliciting gold answers, so we believe these unexpected outcomes alone are a meaningful finding to see the status quo of the models.
>
> #### Q1)
> In the sections denoted 3.2 and 5.3 of our study, we introduced a comprehensive analysis of the reasoning process employed by the model. We wish to clarify a critical point: While we indeed showcased the model's use of references to established research sources like Pew Research and statistical data from credible entities such as the Bureau of Labor Statistics as justifications for certain ratings, it is important to discern that this does not necessarily equate to the model unconditionally 'relying' on these findings.
>
> We acknowledge the subtle but important distinction here. The utilization of such references can be perceived as a logical progression in the model's response process, rather than a strict dependence on these sources. Our intention is not to assert that the model fundamentally 'relied' on these external references, which could inadvertently imply a level of autonomy beyond its capabilities. Thanks for your meticulous attention. We will make sure to clarify this in our manuscript.
>
> #### Q2)
> Yes, we envision ample potential for expanding the current approach across various dimensions. This framework’s versatility seems promising for applications in diverse social groups, personas (as proposed in recent works [1] and [2]), and alternative models, including their future updates. Another intriguing avenue within this framework pertains to investigating model reasoning and explanation.
>
> [1] [Marked Personas: Using Natural Language Prompts to Measure Stereotypes in Language Models](https://aclanthology.org/2023.acl-long.84) (Cheng et al., ACL 2023)
> [2] Perez, E., Ringer, S., Lukošiūtė, K., Nguyen, K., Chen, E., Heiner, S., ... & Kaplan, J. (2022). Discovering language model behaviors with model-written evaluations. arXiv preprint arXiv:2212.09251.
>
> #### Q3)
> For each social group and trait combination, we performed 10 rounds of generations. This meticulous iteration was conducted to establish a robust foundation for our analysis and ensure a comprehensive grasp of the model's behavior.
>
> To further gauge the reliability and consistency of the generated ratings across these rounds, we also calculated the coefficient of variation (CV) conditioned on each social demographic group, represented as the ratio of the standard deviation (σ) to the mean (μ) ($\frac{\sigma}{\mu}$). This calculation quantifies the extent of variability relative to the mean of the population.
>
> | Model                   | Davinci      | GPT-3.5     | BARD         |   |   |   |   |   |   |
> |-------------------------|--------------|-------------|--------------|---|---|---|---|---|---|
> | Competence CV Avg (Max) | .069 (0.338) | .043 (.296) | .101 (.384)  |   |   |   |   |   |   |
> | Warmth CV Avg (Max)     | .061 (0.167) | .032 (.250) | .082 (.266)  |   |   |   |   |   |   |
>
> A score of 0 indicates perfect consistency over the generations, while a larger scale indicates less consistency. We notice that overall CV scores are lower than 1 and the maximum of CV scores is also less than 0.5.
>
> #### Q4)
> Thanks for the comment. The values show the average refuse-to-answer rate and we clarify that the graph shows the top 10 social demographic groups. We will make sure to make detailed explanations in the manuscript. We ran 10 rounds of generations for all groups. The values show the average refuse-to-answer rate.
>
> ####
> Comments on Reproducibility: We will make sure that the code sources that are made to generate and analyze responses will be made available for reproducing the results.
>
> ####
> Once again, we thank the reviewers for taking their time to read through, and their thoughtful comments. Please let us know if you have any follow-up questions to discuss. Thanks!

---

### Official Review · Reviewer_aZZz · 2023-07-31

**Soundness:** 4

**Excitement:**

4: Strong: This paper deepens the understanding of some phenomenon or lowers the barriers to an existing research direction.

**Missing References:**

Understanding and Countering Stereotypes: A Computational Approach to the Stereotype Content Model. Fraser et al. ACL 2021. https://aclanthology.org/2021.acl-long.50/


**Paper Topic And Main Contributions:**

The authors propose a framework called StereoMap for “understanding” stereotypes encoded in LLMs. Their framework is inspired by the stereotype-content model (SCM) from psychology research and measures two dimensions of social groups: warmth (e.g. morality, sociability) and competence (e.g. ability, assertiveness). The authors then apply their framework onto several LLMs (Google Bard and OpenAI models) to see how much these model’s outputs align with known stereotypes in psychology literature (Figure 2).

**Questions For The Authors:**

The authors use a warmth-competence dictionary to analyze keywords provided by LLMs. What percentage of generated keywords were covered by this dictionary? What do you do when a generated keyword is not in this dictionary?


**Reasons To Accept:**

This paper is very well-written and I enjoyed reading it. It hits all of the key points that one may expect from a paper that introduces the use of SCM for LLMs. The NLP community needs more work that integrates established and well-known frameworks from psychology (such as SCM) into analysis of LLMs.

**Reasons To Reject:**

The prompts used in this paper directly ask models to rate warmth and competence in structured response formats, but it’s likely that warmth and competence associated with social groups can emerge in more flexible or subtle downstream tasks, e.g. in free-form generated narratives (e.g. Lucy & Bamman 2021’s GPT-3 & gender bias paper) or descriptions (e.g. Cheng et al. ACL 2023’s “marked personas” paper). However, I don’t think the narrowness of the prompts tested in paper is strong enough weakness to warrant rejection, as additional prompting styles and response analyses that incorporate SCM could instead be part of a future line of work initiated by this current paper.

One downside is that the models used by this paper are closed-source ones that may change in unpredictable and non-transparent ways. Again this is a small weakness as the main contribution of this paper is a framework for measuring stereotypes and a demonstration of that framework rather than just the LLM-specific findings themselves.

Text generated by a model is not necessarily indicative of the actual underlying reasoning or explanation of a model, so I might caution the wording the authors use in Section 5.3.


**Reproducibility:**

3: Could reproduce the results with some difficulty. The settings of parameters are underspecified or subjectively determined; the training/evaluation data are not widely available.

**Reviewer Confidence:**

4: Quite sure. I tried to check the important points carefully. It's unlikely, though conceivable, that I missed something that should affect my ratings.

**Typos Grammar Style And Presentation Improvements:**

Table 1’s caption says “A sample prompt and response” but actually shows three prompts and responses.

I understand that some information is in the Appendix for space reasons, but in line 294 it would be good to summarize what kinds of social groups are tested in this paper.

Understanding Figure 3 requires scrolling back and forth between it and Figure 2 to figure out what groups the cluster numbers correspond to. (Unfortunately I don’t have satisfying solutions for this problem, but wanted to bring it up in case the authors wish to reconsider figure design.)

---

> ### Author Rebuttal · Authors · 2023-08-28
>
> We’d like to thank the reviewer for your comments and for the time and effort you put into reading, understanding, and evaluating our paper. In particular, we highly appreciate your suggestions and questions to make the writing clear. We will make sure to clarify the points along with the suggestions on the presentation improvements. As you mentioned, we envision this work can be extended or converged with various downstream tasks, or extended to other open-sourced models as well.
>
> - Typos in Table 1: Thanks for pointing this out we will clarify this as ‘A sample -> Samples’. We will also add the summary of the social groups tested in this work in the main text.
> - Figures in the Appendix: We also had this concern in presenting our paper. We are figuring out an efficient way to present these results in our final presentation.
> - The comment in section 5.3. Yes, we totally agree with you. We clarify that the reasoning does not equate with the actual reasoning behind the models’ judgment.
>
> Regarding the coverage ratio, we initially calculated coverage (row1, Initial Coverage), and to enhance it, we employed lemmatization on generated words, using the spacy en_core_web_sm lemmatizer. This led to increased coverage, as shown in row 2. For the remaining uncovered words, a manual inspection revealed instances where two-word phrases failed to be lemmatized (e.g., ‘risk takers’, ‘risk taking’ to ‘risk taker’). We manually mapped those cases which ended up with the coverage noted in row 3.
>
> While an option to increase coverage existed, we weighed the advantage of leveraging the given dictionary against manually annotating to ensure precision and validity. Thus, the words not covered in the dictionary were not considered in our analysis, and row 3 was deemed the optimal version for reporting results.
>
> |                               | Davinci | GPT-3.5 | BARD    |   |   |   |   |   |   |
> |-------------------------------|---------|---------|---------|---|---|---|---|---|---|
> | Initial Coverage (%)          | 65.40%  | 67.80%  | 61.30%  |   |   |   |   |   |   |
> | +Lemmentization Coverage  (%) | 70.5%   | 74.40%  | 67.00%  |   |   |   |   |   |   |
> | +Coverage (%)                 | 75.3%   | 76.40%  | 72.60%  |   |   |   |   |   |   |
>
> Once again, we thank the reviewer for your valuable comments, and we are more than happy to engage in discussions if you have any follow-up questions. Thanks.

---

### Official Review · Reviewer_NHSL · 2023-08-04

**Soundness:** 3

**Excitement:**

3: Ambivalent: It has merits (e.g., it reports state-of-the-art results, the idea is nice), but there are key weaknesses (e.g., it describes incremental work), and it can significantly benefit from another round of revision. However, I won't object to accepting it if my co-reviewers champion it.

**Missing References:**

Understanding Stereotypes in Language Models: Towards Robust Measurement and Zero-Shot Debiasing

StereoSet: Measuring stereotypical bias in pretrained language models

**Paper Topic And Main Contributions:**

The paper presents an approach to understanding how LLMs perceive and represent social groups, with a particular focus on identifying and analyzing harmful stereotypes encoded in their outputs. The contribution of the paper is the introduction of the STEREOMAP framework, which is grounded in the well-established Stereotype Content Model (SCM) from psychology.

**Questions For The Authors:**

Given the potential impact of the prompt design on the LLMs' responses, did you conduct any pilot studies or validation procedures to assess the effectiveness and appropriateness of the prompts before implementing them in the main study

How did you ensure that the prompts were structured to allow the LLMs to provide comprehensive and informative responses regarding the reasons behind their perceptions of social groups?

**Reasons To Accept:**

The authors' decision to incorporate established psychological theories (SCM) to measure stereotypes in LLMs is well-justified. By doing so, the paper addresses concerns related to construct validity and provides a solid theoretical foundation to identify and measure stereotypes in LLM outputs more accurately.

**Reasons To Reject:**

The study could potentially have weaknesses in experimental design, such the selection of prompts, which might impact the validity and reliability of the results. It's essential to ensure that prompts are neutral, and do not lead respondents in any particular direction.

**Reproducibility:**

4: Could mostly reproduce the results, but there may be some variation because of sample variance or minor variations in their interpretation of the protocol or method.

**Reviewer Confidence:**

4: Quite sure. I tried to check the important points carefully. It's unlikely, though conceivable, that I missed something that should affect my ratings.

---

> ### Author Rebuttal · Authors · 2023-08-28
>
> We genuinely appreciate your insightful comments and queries. We acknowledge and understand the concerns you’ve raised, and we’re pleased to provide clarifications on our methodological choices.
>
> The main objective of this research was to examine the large language models’ (LLMs) behavior. To this end, we adopted an experimental design that has been well grounded in theory and that has been well validated. Thus, the fundamental foundation and configuration of our prompts were, in essence, drawn from a prior study [1],[2]. These prompts represent the questions that were specifically designed to gauge stereotypes among human subjects, incorporating a meticulous construct validity approach. This informed our decision to adopt these questions as the cornerstone of our research. By utilizing these questions as our baseline, we sought to establish a meaningful parallel between the behavior (responses) of LLMs and human responses within a controlled context.
>
> Building on this chosen foundation, we introduced subtle refinements to create the final prompts, including the incorporation of a Chain of Thoughts (CoT) style prompt.  Below, we present an ablation analysis on the variations of the prompts: prompts without CoT (w/o CoT), prompts without Instructions (w/o Instructions), and prompts without CoT and Instructions (w/o CoT+Instructions). We ran an extra 5 rounds of each prompt style per every social demographic group considered. The metrics we adopted are the coefficient of Variation (CV), the variability relative to the mean conditioned on each social demographic group (represented as the ratio of the standard deviation (σ) to the mean (μ) ($\frac{\sigma}{\mu}$) as indicators of score consistency, Self-BLEU [3], and Entropy over n-gram distribution [4], as indicators of the diversity in reasoning. We also calculated the Refuse to Answer ratio given each style of prompt.
>
> |                                | Score Consistency       |                     | Reasoning Diversity |              |              | Refuse to Answer Ratio (%)  |   |   |   |
> |--------------------------------|-------------------------|---------------------|---------------------|--------------|--------------|-----------------------------|---|---|---|
> | Prompt Type                    | Competence CV Avg (max) | Warmth CV Avg (max) | Self-BLEU (↓)       | Entropy-2(↑) | Entropy-3(↑) |                             |   |   |   |
> | Prompt (paper)                 | .043 (.296)             | .032 (.250)         | 0.064               | 12.02        | 16.28        | 0.01                        |   |   |   |
> |         - w/o CoT              | .041 (.279)             | .048 (.263)         | 0.065               | 10.77        | 14.41        | 0.053                       |   |   |   |
> |         - w/o Instructions     | .042 (.165)             | .031 (.093)         | 0.062               | 10.83        | 14.49        | 0.057                       |   |   |   |
> |         - w/o CoT+Instructions | .040 (.279)             | .046 (.263)         | 0.065               | 10.85        | 14.51        | 0.055                       |   |   |   |
>
> The results show the marginal differences between prompt styles, with the mean score consistency ranging from 0.03 to 0.04. (We also calculated the consistency CV between the chosen prompt and the prompt w/o CoT, w/o Instruction, and w/o CoT+instructions, giving CV avg of 0.093, 0.094, and 0.093 respectively, which indicates high consistency across prompt types.) The reasoning diversity scores also showed marginal differences, with the Self-BLEU score ranging from 0.062 to 0.065.  Along with the metrics, we manually had a look at the model-generated responses for each style and chose the prompt that had the lowest refuse-to-answer ratio given diverse reasoning scores in presenting our results.
>
> To assess the given framework and model responses, we additionally conducted human evaluations. We sampled 5 responses per every social demographic group across 3 models - Davinci, Gpt-3.5, and BARD. We obtained the assessments through 3 individuals participating in the evaluation process. Criteria considered for human evaluation are:
> - Relevance (between Scores and Reasoning): The alignment between the scores assigned by the models and the reasoning they provided. This assessment discerns the congruence between the generated ratings and the accompanying rationale. The score scale ranged between 0 (not relevant) to 1 (relevant)
> - Soundness (between Keywords and Reasoning): This criterion scrutinized whether the keywords employed by the models were in harmony with the provided reasoning. The scale ranged between 0 (no sound) to 1 (sound)
> - Coherence: Given the iterative nature of our evaluation, we placed emphasis on coherence, investigating if the models consistently generated coherent outputs across multiple instances. The scores ranged from 0 (not coherent) to 1 (coherent)
>
> |                      | DAVINCI      | GPT-3.5      | BARD          |   |   |   |   |   |   |
> |----------------------|--------------|--------------|---------------|---|---|---|---|---|---|
> | Relevance Mean (std) | 0.938 (0.23) | 0.945(0.22)  | 0.942 (0.23)  |   |   |   |   |   |   |
> | Soundness Mean (std) | 0.931 (0.25) | 0.945 (0.22) | 0.897 (.30)   |   |   |   |   |   |   |
> | Coherence Mean (std) | 0.948 (0.22) | 0.959 (0.19) | 0.918 (0.27)  |   |   |   |   |   |   |
>
> We’d like to note that these evaluation criteria are not to discern whether the scores or model responses are right or wrong, but rather to assess the coherence, and soundness of models’ outputs which mainly focus on the framework evaluation. The scores obtained in human evaluation were fairly high, around 0.8 to 0.9. Given these scores, however, we acknowledge that one might still question the validity of the suggested prompting method. As we do not have access to the oracle, the ground truth of what LLMs really know, we’d like to emphasize our work as an attempt to probe LLMs’ behavior through empirical experiments. Under this objective, we endeavored as much as possible to ground the experiment's design mainly on the previous theories and questions validated by social science studies. Also, we have conducted additional analysis on the score consistency along with other metrics to show the choice of our method.
>
> Once again, we thank you for your thoughtful consideration and questions and we hope the comments have answered your concerns. Please feel free to reach out if you wish to discuss. We hope to engage in follow-up discussions through open review!
>
> [1] Susan T Fiske, Amy JC Cuddy, Peter Glick, and Jun Xu. 2002. A model of (often mixed) stereotype content: competence and warmth respectively follow from perceived status and competition. Journal of personality and social psychology, 82(6):878.
>
> [2] Cuddy, A. J., Fiske, S. T., & Glick, P. (2007). The BIAS map: behaviors from intergroup affect and stereotypes. Journal of personality and social psychology, 92(4), 631.
>
> [3] Zhu, Y., Lu, S., Zheng, L., Guo, J., Zhang, W., Wang, J., & Yu, Y. (2018, June). Texygen: A benchmarking platform for text generation models. In The 41st international ACM SIGIR conference on research & development in information retrieval (pp. 1097-1100).
>
> [4] Zhang, Y., Galley, M., Gao, J., Gan, Z., Li, X., Brockett, C., & Dolan, B. (2018). Generating informative and diverse conversational responses via adversarial information maximization. Advances in Neural Information Processing Systems, 31.

---

### Meta-Review · Area_Chair_LwCS · 2023-09-15

**Recommendation:** 4

**Metareview:**

This paper examines the extent to which knowledge of social stereotypes is encoded in LLMs, by prompting for evaluations of various social groups along salient dimensions (e.g., warmth and competence). The dimensions, as well as the prompts, are grounded in social science theory, namely Fiske’s Stereotype Content Model (SCM).  Reviewers appreciated the fact that the study was interdisciplinary and based in social science findings, and are generally in agreement that the methodology is sound (ratings of 3-4). There were some concerns about how dependent the results are on the specific prompts used, but this is addressed quite conclusively by the authors in their rebuttal with further discussion as well as an ablation study. Additional discussion on cases where the LLM results diverge from expected results (i.e. published human studies), as well as how stereotypical biases can be more subtly expressed in open-ended text responses, would strengthen the revised version.

While two reviewers are “ambivalent” with respect to excitement, one reviewer is “strongly” excited by the work.

AC Note: It was not raised in the reviews, but I would caution against attributing these stereotypes to the LLMs directly. The prompts specifically state “I am not interested in your personal beliefs, but in how you think they are viewed by others.” Therefore, in statements like “We propose a framework StereoMap that measures *large language models' perceptions of social groups* using dimensions of Warmth and Competence” I would prefer to see something like “measures *large language models’ perceptions of how social groups are viewed in society*”.  As an example of this distinction, I do not personally believe that women are irrational or can’t do math, but I have knowledge that these exist as stereotypes, which is what the question asks about.

Pros:
- Interesting study examining what knowledge of social stereotypes is encoded in LLMs
- Grounded in Fiske’s Stereotype Content Model from social psychology
- Well-written paper

Cons:
- Potential disconnect between model’s stated stereotype “beliefs” when asked directly, versus more subtle expressions of stereotypical bias that can occur in open-ended tasks
- Possible dependence of results on specific prompts

---

### Decision · Program_Chairs · 2023-10-07

**Decision:**

Accept-Main

**Comment:**

This paper examines the extent to which knowledge of social stereotypes is encoded in LLMs, by prompting for evaluations of various social groups along salient dimensions (e.g., warmth and competence). The dimensions, as well as the prompts, are grounded in social science theory, namely Fiske’s Stereotype Content Model (SCM).  Reviewers appreciated the fact that the study was interdisciplinary and based in social science findings, and are generally in agreement that the methodology is sound (ratings of 3-4). There were some concerns about how dependent the results are on the specific prompts used, but this is addressed quite conclusively by the authors in their rebuttal with further discussion as well as an ablation study. Additional discussion on cases where the LLM results diverge from expected results (i.e. published human studies), as well as how stereotypical biases can be more subtly expressed in open-ended text responses, would strengthen the revised version.

While two reviewers are “ambivalent” with respect to excitement, one reviewer is “strongly” excited by the work.

AC Note: It was not raised in the reviews, but I would caution against attributing these stereotypes to the LLMs directly. The prompts specifically state “I am not interested in your personal beliefs, but in how you think they are viewed by others.” Therefore, in statements like “We propose a framework StereoMap that measures *large language models' perceptions of social groups* using dimensions of Warmth and Competence” I would prefer to see something like “measures *large language models’ perceptions of how social groups are viewed in society*”.  As an example of this distinction, I do not personally believe that women are irrational or can’t do math, but I have knowledge that these exist as stereotypes, which is what the question asks about.

Pros:
- Interesting study examining what knowledge of social stereotypes is encoded in LLMs
- Grounded in Fiske’s Stereotype Content Model from social psychology
- Well-written paper

Cons:
- Potential disconnect between model’s stated stereotype “beliefs” when asked directly, versus more subtle expressions of stereotypical bias that can occur in open-ended tasks
- Possible dependence of results on specific prompts